# Can LLMs Enhance Performance Prediction for Deep Learning Models?

**Karthick Panner Selvam** [1]   **Phitchaya Mangpo Phothilimthana** [2]   **Sami Abu-El-Haija** [3]   **Bryan Perozzi** [3]
**Mats Brorsson** [1]

## Abstract

Accurate performance prediction of Deep Learning (DL) models is essential for efficient resource allocation and optimizations in various stages of the DL system stack. While existing approaches can achieve high prediction accuracy, they lack ability to quickly adapt to new hardware environments or emerging workloads. This paper leverages both Graph Neural Networks (GNNs) and Large Language Models (LLMs) to enhance the accuracy and adaptability of DL performance prediction. Our intuition is that GNNs are adept at capturing the structural information of DL models, naturally represented as graphs, while LLMs provide generalization and the ability to quickly adapt to various tasks thanks to extensive pre-training data. We empirically demonstrate that using GNN-derived graph embeddings as inputs to an LLM outperforms traditional representations, including high-level text summary and lossless semi-structured text (e.g., JSON), for this task. Furthermore, we propose a structured pre-training strategy to enable model adaptation to new hardware environments, significantly reducing the need for extensive retraining. Our experiments validate the effectiveness of this approach, showing an 8.8 percentage-point improvement in accuracy over a state-of-the-art GNN baseline. Notably, when adapted to new hardware with few samples, our method achieves a remarkable 30–70 percentage-point increase in accuracy compared to the GNN baseline.

[1]University of Luxembourg [2]Google DeepMind [3]Google Research. Correspondence to: Karthick Panner Selvam <karthick.pannerselvam@uni.lu>.

Accepted to the Workshop on Advancing Neural Network Training at International Conference on Machine Learning (WANT@ICML 2024).

## 1. Introduction

Performance prediction for Deep Learning (DL) models is essential for all sorts of optimization methods in the DL system stack: from Neural Architecture Search, to model partitioning and sharding, to low-level compiler optimizations. Performance prediction involves estimating various operational metrics — such as inference time, memory usage, and power consumption — that are crucial for efficient hardware utilization and scheduling. Since DL models are computation graphs, researchers have employed Graph Neural Networks (GNNs) to extract information from the DL model for various optimization decisions given hardware components (Phothilimthana et al., 2023; Panner Selvam & Brorsson; 2023; Liu et al., 2022).

Unfortunately, aforementioned GNN-based approaches require comprehensive retraining to accommodate new hardware environments or DL architectures, often requiring large labeled datasets. These requirements can hinder a rapid adaptation and optimization, limiting the flexibility of these models when new architectures or configurations emerge. Fortunately, the recent successes of Large Language Models (LLMs) in various domains have underscored their capability to understand and generate complex systems (Team et al., 2024; Singhal et al., 2023; Wayne et al., 2023; Wu et al., 2023; Li et al., 2023). This includes not only natural language but also structured data such as code, configuration settings, and textual descriptions of hardware configurations and DL architectures. Given their extensive pre-training on diverse datasets, LLMs can generalize effectively when fine-tuned on specific tasks. Their generalization capability makes them good candidates for enhancing DL performance prediction.

However, employing LLMs in the performance prediction domain poses challenges, primarily due to the need for representing DL models in a format that LLMs can efficiently process. Prior works have considered using high-level descriptions to represent programs and graphs as text inputs for compiler optimizations and performance predictions (Cummins et al., 2023; Jawahar et al., 2023). Nonetheless, these representations often fail to maintain the full structural intricacies of DL models, losing crucial connectivity and hierarchical information. An alternative representation is

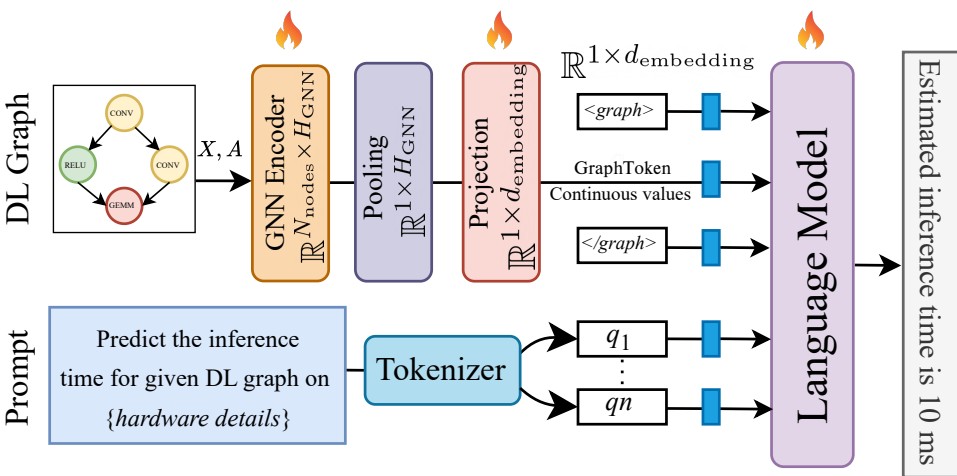

Figure 1: Our approach integrates GNNs and LLMs for DL model performance prediction. The methodology utilizes soft prompting to fine-tune pre-trained GNN weights and projection layer weights, while updating pre-trained LLMs with the LoRA technique. During fine-tuning, gradients flow from the LLM to the GNN, allowing the system to process graphs and prompts effectively and generate accurate performance metrics predictions.

to use structured text format (e.g. JSON, XML, Protobuf, etc.), which maintains detailed information of node features and their connections. However, DL models can contain tens-of-thousands of nodes (Phothilimthana et al., 2023), which can hinder the processing efficiency and scalability when used with LLMs.

Recent research has explored the use of GNNs as encoders to convert graph data into embeddings as inputs to LLMs, thereby effectively bridging the gap between graph data and the textual input preferred by LLMs. However, these studies primarily focus on graph-based question answering, rather than directly on performance prediction (Perozzi et al., 2024; Liu et al., 2024).

In line with (Perozzi et al., 2024) findings, We hypothesize that graph embeddings, derived from GNNs, represent DL models more effectively for performance prediction than conventional text representations because the graph embeddings could better capture structures and connectivity. Based on this hypothesis, we propose the GNN-LLM model for DL performance prediction, as illustrated in Figure 1. Our experiment has confirmed that using graph embeddings significantly outperforms using a semi-structured text format (JSON) and a high-level text format in both accuracy and computational efficiency. Specifically, our approach surpasses a JSON format by approximately 6% in accuracy and is 21 times faster in terms of training time. Likewise, our approach surpasses high-level text by 134% in accuracy and is 2 times faster in terms of training time, demonstrating a substantial improvement over text-based representation.

To enhance the adaptability and accuracy of the model, we further develop a structured pre-training strategy that ob-

viates the need for extensive retraining from scratch. The approach begins by training a GNN using a mask autoencoding technique on unlabeled DL models, inspired by (Hou et al., 2022) research. In this initial phase, the GNN learns to capture DL graph structures and node information. Subsequently, we refine the integration between the DL graph data and the LLM by fine-tuning the projection layer and the LLM through a graph-to-text task. This graph-to-text translation will enable the LLM to comprehend DL graph structures and improve the model's ability to adapt to new hardware with minimal training samples for downstream performance prediction tasks. Finally, all components are fine-tuned for the final performance prediction task.

In the evaluation, our method achieves a 8.8 percentage-point increase in accuracy over the state-of-the-art GNN baseline on the NNLQP multi-platform dataset, and a remarkable 30–70 percentage-point increase in accuracy when adapted to new hardware with few samples. The results confirms our method's efficacy in enhancing both the accuracy and adaptability of performance predictions across varied computational environments.

1. We empirically evaluate different DL model representations for LLMs on performance prediction tasks, showing that a graph embedding-based input is most effective.

2. We introduce a method integrating GNNs and LLMs for the DL performance prediction domain, combining GNNs' structural insights and LLMs' generalization capabilities.

3. We propose a structured pre-training strategy to en-

hance model performance in a new hardware environment with limited training samples.

4. We contribute a specialized graph-to-text dataset designed to further research into the integration of GNN and LLMs. This dataset is particularly valuable for benchmarking and advancing the application of GNN-LLM combinations in graph learning tasks.

5. Our research offers a promising direction for improving DL performance prediction accuracy and adaptability across diverse hardware environments.

## 2. Related Work

The field of performance prediction for DL models has witnessed growing interest in recent years. Early work by (Qi et al., 2017) proposed an analytical model to estimate DL model training time. Subsequent studies like that of (Gao et al., 2020) extended these methods to predict memory consumption, utilizing analytical models to estimate resource utilization during training. To improve prediction accuracy, researchers have explored machine learning approaches. (Bouhali et al., 2021) used an MLP-based regressor with features like trainable parameter counts, but it was constrained by a shallow understanding of DL layers' dynamics. Others, such as (Justus et al., 2018) and (Gianniti et al., 2018), adopted a layer-by-layer technique, they predicted performance for each layer instead of the whole model, incorporating parameters like FLOPs to predict execution times and power consumption.

However, this layerwise strategy failed to capture the network structure of DL models (Liu et al., 2022) To address this limitation, many methods (Kaufman et al., 2021; Dudziak et al., 2020; Liu et al., 2022; Bai et al., 2022; Yi et al., 2023; Zhou et al., 2020; Phothilimthana et al., 2023; Panner Selvam & Brorsson, 2023; Panner Selvam & Brorsson) utilized graph learning techniques to generate embeddings that encapsulate the DL model network topology, as well as the features of the computation graph. These embeddings are trained to predict performance characteristics.

Despite these advancements, prior approaches lack online adaptability. Current methods require retraining for new DL architectures or hardware configurations. On the other hand, our proposed approach aims to overcome this challenge by integrating GNNs with LLMs to create a predictive system that is more adaptable and flexible in real-world scenarios.

## 3. Background

### 3.1. DL Models as Computational Graphs & Graph Neural Networks (GNNs)

DL models can be represented as directed acyclic computational graphs, where nodes correspond to mathematical operations and edges represent data flow between these operations. The input features of every node include the *op code* (*e.g.*, einsum, relu, *etc*), the output data type (*e.g.*, float32, uint8, *etc*) and the shape of the output tensor – See (Phothilimthana et al., 2023) for comprehensive list of node-wise features.

GNNs are designed to operate on graph-structured data. Let graph of $n$ nodes be represented with a node feature matrix $\mathbf{X} \in \mathbb{R}^{N \times \cdot}$ and an adjacency matrix $\mathbf{A} \in \{0, 1\}^{N \times N}$. GNNs use an iterative message passing process to generate embeddings for nodes. During message passing, each node updates its embedding by aggregating information from its neighbors. GNN layer can be written as:

$$\mathbf{H}^{(l)} = \text{TRANSFORM}\left(\mathbf{H}_i^{(l-1)}, \mathbf{A}\right) \qquad (1)$$

where $\mathbf{H}^{(l)}$ is the node embedding matrix at the $l$-th layer and $\mathbf{H}^{(l)} = \mathbf{X}$. Through multiple message-passing layers, each node aggregates information from a wider neighborhood, capturing both immediate and distant neighbor information. GNNs have excelled in tasks such as node classification, link prediction, and graph-level classification. There are many possible choices for TRANSFORM function (Kipf & Welling, 2017; Veličković et al., 2018; Hamilton et al., 2017; Xu et al., 2019). In our work, we use a variant of the GIN model (Xu et al., 2019):

$$\mathbf{H}_{\text{GIN}}^{(l)} = \text{MLP}\left((\mathbf{A} + \mathbf{I}\epsilon)\,\mathbf{H}^{(l-1)}\right), \qquad (2)$$

where MLP stands for multi-layer perceptron, $\mathbf{I}$ is $n \times n$ identity matrix, and $\epsilon$ is small constant.

### 3.2. Large Language Models

#### 3.2.1. PRE-TRAINED LARGE LANGAUGE MODELS

Pre-trained LLMs are advanced neural networks for natural language processing tasks. They leverage the Transformer architecture (Vaswani et al., 2017), which uses self-attention mechanisms to manage long-range dependencies in text. LLMs are pre-trained on extensive corpora to predict subsequent tokens, enabling them to capture intricate linguistic patterns. This pre-training is followed by finetuning task-specific datasets to adapt to various applications like text classification and translation.

#### 3.2.2. PARAMETER-EFFICIENT FINE-TUNING

With the rapid increase in the size of state-of-the-art LLMs, traditional fine-tuning has become resource intensive. Parameter-Efficient Fine-Tuning (PEFT) aims to adapt models to new tasks by updating only a small subset of parameters (Xu et al., 2023).

**Low-Rank Adaptation (LoRA)**: LoRA introduces low-rank matrices into model layers, represented as $\Delta W = BA$,

where $B$ and $A$ are trainable low-rank matrices. This approach reduces the computational burden by updating fewer parameters while keeping the main model's parameters frozen, thus preserving the pre-trained knowledge (Hu et al., 2021).

**Soft Prompting**: Soft prompts are learnable vectors integrated into the model's input to guide its behavior toward specific tasks. This method updates only a small number of parameters, making it computationally efficient and preserving the broad knowledge of the model(Bulat & Tzimiropoulos, 2023).

# 4. Methodology

Figure 1 displays our proposed model architecture. Our approach takes a DL model graph and a textual prompt as inputs. The DL graph is initially processed by a GNN encoder and then projected as an embedding to an LLM, along with the token embeddings of the textual prompt.

## 4.1. DL Representation

We consider the following methods to represent DL models for processing by LLMs.

**Graph Representation.** This method first encodes a DL model in the Open Neural Network Exchange (ONNX) format, represented as a graph with node feature matrix $\mathbf{X}$ formulated as:

$$\mathbf{X}_v = \mathbf{X}_v^{(\text{op})} \oplus \mathbf{X}_v^{(\text{attr})} \oplus \mathbf{X}_v^{(\text{shape})} \quad \forall v \leq N \qquad (3)$$

where $\mathbf{X}_v^{(\text{op})}$ is the one-hot encoded vector indicating the type of the node operation. $\mathbf{X}_v^{(\text{attr})}$ includes the node's attribute vector, containing parameters such as kernel size and stride, and $\mathbf{X}_v^{(\text{shape})}$ encodes the output shape. The operation $\oplus$ represents a vector concatenation. This method is adapted from the framework established in (Liu et al., 2022). Subsequently, we feed the node feature matrix $\mathbf{X}$ and the adjacency matrix $\mathbf{A}$ into the GNN. The GNN then produces a graph embedding for input into the LLM, along with prompt's token embeddings, to predict the model's performance.

**High-level Text Representation.** We use a predefined template that captures essential computational and structural properties of a DL model. This includes overall model statistics — such as FLOPs, parameter count, and batch size — offering insights into the model's complexity and capacity. We also include layer-specific statistics, detailing each layer's FLOPs and parameter counts. These elements together offer a holistic view of a DL model's architecture and its computational behavior. To predict its performance, we simply tokenize and apply a conventional word encoder

on the textual prompt for LLM processing.

**Semi-Structured Text Representation.** We adopt a semi-structured JSON format to comprehensively encapsulate a DL architecture. This format itemizes each node's characteristics, including the operator type, input and output shapes, computation complexities, and node attributes. Additionally, it capture node connectivity. For LLM processing, we tokenize and apply a conventional word encoder on the semi-structured description.

## 4.2. Graph Encoding

Our GNN encoder is based on the Graph Isomorphism Network (GIN)(Xu et al., 2019), defined as:

$$\mathbf{H}_{\text{ours}}^{(l)} = (\mathbf{A} + \mathbf{I}\epsilon)\,\text{MLP}(\mathbf{H}^{(l-1)})$$
$$\text{with MLP}(\mathbf{Z}) = \text{ReLU}(\text{BN}(\mathbf{Z}\mathbf{W}_1 + \mathbf{b}_1))\mathbf{W}_2 + \mathbf{b}_2 \qquad (4)$$

We inspired this architecture by (Hou et al., 2022). This setup ensures each node feature undergoes transformation, normalization, and activation, promoting the learning of non-linear dependencies. After processing through $L$ layers, we aggregate node features to form a graph-level representation:

$$\mathbf{g} = \frac{1}{N} \sum_{i=1}^{N} \mathbf{H}_i^{(L)}. \qquad (5)$$

Next, the projection layer transforms the GNN output $\mathbf{g}$ into an embedding vector of size $d_{\text{embedding}}$ for the LLM processing.:

$$\text{GraphToken} = \text{MLP}_{\text{proj}}(\mathbf{g}), \qquad (6)$$

where $\text{MLP}_{\text{proj}}$ encapsulates a series of linear transformations and non-linear activations. It ensures the alignment of dimensionalities and contextual relevance. Note that the output dimension size of the projection is larger than the input dimension size: $|\text{GraphToken}| > |\mathbf{g}|$.

The LLM input is then constructed by integrating graph embeddings GraphToken with token embeddings $\mathbf{Q}$. A textual prompt describing the task like "`Predict the inference time of DL model`" is tokenized as $\mathbf{q} = [q_1, q_2, \ldots, q_n]$. The tokens are then converted into word embeddings: $\mathbf{Q} = \mathbf{E}[\mathbf{q}]$, where $\mathbf{E}$ represents the embedding matrix. The complete LLM input is the concatenation of the projected graph embedding and the token embeddings:

$$\text{Input}_{\text{LLM}} = [\text{<graph>}, \text{GraphToken}, \text{</graph>}, \mathbf{Q}]. \qquad (7)$$

In this sequence, <graph> and </graph> are text tokens directly generated by the tokenizer, marking the beginning and end of the graph embedding. A single graph embedding vector GraphToken efficiently encapsulates the entire graph's structure, compactly representing complex information in a form that complements textual embeddings in LLMs.

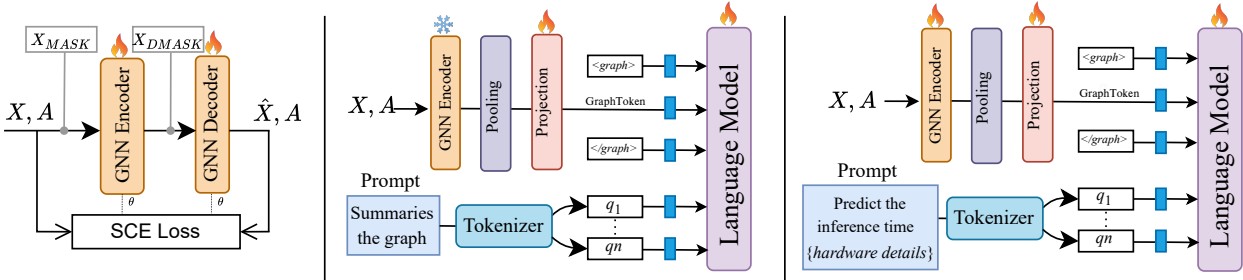

Figure 2: The three stages of our approach: (1) **GNN Pre-training**: Using Scaled Cosine Error (SCE) loss with masked node features ($X_{\text{MASK}}$) approach to pre-train the GNN. (2) **Graph-Text Adaptation**: Fine-tuning the pre-trained GNN encoder (frozen) and updating LLM weights and projection weights using soft prompting and LoRA techniques. (3) **Performance Prediction Fine-tuning**: Updating all GNN projection and LLM parameters through soft prompting and LoRA techniques to predict performance metrics for deep learning graphs on various hardware.

### 4.3. Training Strategy: 3-stage training

We hypothesize that directly fine-tuning both LLMs and GNNs for performance prediction tasks, starting from scratch, may not yield optimal adaptability for new tasks. The challenge lies in the initial lack of domain-specific knowledge, which is crucial for the model to effectively process and predict the DL performance metrics. To address this, we propose a novel structured pre-training methodology, designed to enhance the model's intrinsic understanding of DL graph structures before fine-tuning for performance prediction. The pre-training strategy comprises the three stages as shown in Figure 2.

**Stage 1. GNN Pre-training.** We employ the Graph Maked Auto Encoder technique (GraphMAE) for GNN pre-training (Hou et al., 2022). We use GIN as both encoder and decoder. Given a DL graph with $\mathbf{X}$ and $\mathbf{A}$, we mask a portion of $\mathbf{X}$ using a learnable mask vector to produce $\widetilde{\mathbf{X}}$. The GIN encoder processes $(\widetilde{\mathbf{X}}, A)$ to generate latent embeddings $\mathbf{Z}$, effectively capturing the obscured structural details. The GIN decoder reconstructs the node features from $\mathbf{Z}$ to $\widehat{\mathbf{X}}$, aimed at closely approximating the original $\mathbf{X}$. Reconstruction accuracy is quantified using Scaled Cosine Error (SCE), which evaluates alignment in both direction and magnitude of the feature vectors. Using GIN for both encoding and decoding optimizes the preservation and reconstruction of local graph structures, essential for understanding DL graphs. The SCE, by assessing both vector orientation and length, enhances model sensitivity to structural and feature variations, preparing it for robust performance on subsequent tasks.

**Stage 2. Graph-Text Adaptation.** For this stage we update only projection layer and LLM weights. The projection layer $\mathbf{W}_p$ adapts the graph embeddings GraphToken for integration with the LLM. During training, we update the

projection layer weights using soft prompting techniques.

$$\frac{\partial \mathcal{L}}{\partial \mathbf{W}_p} = \frac{\partial \mathcal{L}}{\partial \text{Output}} \cdot \frac{\partial \text{Output}}{\partial \text{GraphToken}} \cdot \text{GraphToken}^T$$

Here, $\frac{\partial \mathcal{L}}{\partial \text{Output}}$ represents the gradient of the loss with respect to the LLM's output, and $\frac{\partial \text{Output}}{\partial \text{GraphToken}}$ captures how changes in GraphToken affect the output. We used cross-entropy loss for the next word prediction. We utilize the LoRA technique to efficiently update the LLM weights. The updates for the low-rank matrices $\mathbf{B}$ and $\mathbf{C}$ are given by:

$$\frac{\partial \mathcal{L}}{\partial \mathbf{B}} = \frac{\partial \mathcal{L}}{\partial \Delta \mathbf{W}} \cdot \mathbf{C}^T, \quad \frac{\partial \mathcal{L}}{\partial \mathbf{C}} = \mathbf{B}^T \cdot \frac{\partial \mathcal{L}}{\partial \Delta \mathbf{W}}$$

where $\Delta \mathbf{W} = \mathbf{BC}$ represents the low-rank update to the LLM weights. The GNN encoder weights remain frozen during this stage to preserve the integrity of the initial graph embeddings learned during pre-training. This selective updating strategy helps maintain foundational graph understanding and ensures consistent model performance across various adaptation scenarios.

**Stage 3. Performance Prediction Fine-Tuning.** In this final stage, we load the pre-trained GIN encoder weights from Stage 1 and the projection $\mathbf{W}_p$ and LoRA weights from Stage 2. We fine-tune the entire GNN to LLM model for performance prediction.

Note that naively feeding the GNN embedding outputs as multiple concrete text tokens to the LLM does not work because the gradient does not flow from the LLM to the GNN. This is why we adopt the proposed approach.

**Training Datasets.** We utilize three distinct datasets for the different stages of our model's training process. For GNN pre-training, we use a dataset containing 20,000 unlabeled DL graphs (Liu et al., 2022). This extensive set of

Table 1: Performance comparison of different representations of DL models for performance prediction tasks. Our proposed method (DL graph as embedding) demonstrates superior accuracy and efficiency, outperforming both JSON and high-level text representations.

| Method | MAPE ↓ | ACC(10%) ↑ | TTT(hr) ↓ | Max Token Length ↓ |
|--------|--------|------------|-----------|--------------------|
| **Ours** | **12.61** | **52.83** | **0.23** | 512 |
| JSON | 13.55 | 49.80 | 5.02 | 2048 |
| Text | 41.42 | 22.50 | 0.46 | 512 |

Table 2: Performance comparison of LLM models with and without graph-text adaptation combined with GNNs having either random or pre-trained weights. Results indicate that graph-text adaptation significantly improves LLM performance.

| Models | MAPE ↓ | Acc (10%) ↑ |
|--------|--------|-------------|
| LLM + GNN | 14.71 | 49.27 |
| LLM + GNN$_{PRE}$ | 20.02 | 36.58 |
| LLM$_{PRE}$ + GNN | 13.57 | 55.12 |
| LLM$_{PRE}$ + GNN$_{PRE}$ | **12.50** | **57.10** |

graphs allows our GNN to capture a wide range of node and edge features, providing robust initial embeddings.

For graph-to-text adaptation, we introduce a novel dataset based on the NNLQP dataset (Liu et al., 2022). Each summary provides comprehensive details, including the total number of nodes, edges, model complexity, and statistics for each layer. The dataset comprises 20,000 prompts.

For performance prediction fine-tuning, we utilize the NNLQP Multi-platform dataset, which encompasses DL graphs, platform IDs, and inference latency metrics across 10 different hardware platforms. The dataset consists of 7,396 graphs designated for training and 3,201 for testing, totaling 10,597 graphs. For additional details on these datasets, please refer to the Appendix A.3.

## 5. Experiments

This section presents a series of experiments designed to validate the efficacy of our integrated model for DL performance prediction. We utilized our performance prediction datasets described in section 4.3 to challenge our model under different conditions and compared it with the GNN baseline to underscore its advantages and unique capabilities. The computing details are explained in appendix A.1.

### 5.1. Experiment: DL Representation

This experiment explores the efficacy of different representations of DL models for performance prediction tasks using LLMs. The DL representations are mentioned in Section 4.1. We investigated three primary formats: our proposed method (DL graph as embedding), semi-structured format (JSON), and high-level text. Each format presents unique challenges in how effectively it can be processed by LLMs.

**Setting**: For this experiment, we utilized the Llama3-8B[1] pre-trained model as the base LLM. The Adam optimizer was used with a learning rate of $1 \times 10^{-5}$, and LoRA with rank 8 was employed for efficient parameter updating. The GIN encoder used a learning rate of $1 \times 10^{-3}$. Each model was trained over 10 epochs, repeated 3 times to ensure stability and convergence of results.

In experiments with the performance prediction dataset (Liu et al., 2022), the entire set of 20,000 ONNX models was converted to JSON format and tokenized using the Llama3-8B model tokenizer to assess context length. The JSON format reached a maximum context length of 18,000 tokens. Therefore, we selected the AlexNet family in the dataset due to its shorter context length compared to other families. A 90:10 train-test split was used for this experiment, consistent with previous work (Liu et al., 2022).

**Result**: Our proposed approach outperforms both JSON and high-level text representations significantly in terms of MAPE and ACC(10%), as shown in Table 1. Our method also demonstrated substantial efficiencies in training time, with the Total Training Time (TTT) notably lower than that required for JSON, which had the highest tokenization length and training duration. These results highlight the critical impact of DL model representation on the performance prediction capabilities of LLMs. High-level text, while simple, fails to capture the necessary connectivity information, leading to poor prediction accuracy. The semi-structured JSON format offers some improvement by providing hierarchical data, but its verbosity and resulting long token sequences increase computational costs. Our proposed method, which embeds the DL graph structure into a compact representation, strikes an optimal balance by preserving essential connectivity information within a manageable token length.

This approach not only enhances prediction accuracy but also ensures computational efficiency. The graph embeddings naturally align with the inherent structure of DL mod-

---

[1]https://llama.meta.com/llama3/

Table 3: Performance comparison of the GNN baseline against our models with different base LLMs (Llama3-8B and Mistral-7B) using a multi-platform performance prediction dataset. Both our models utilize GNN pre-training and graph-text adaptation. The results demonstrate that our approach outperforms the baseline, highlighting the effectiveness of the integrated method.

| Platforms | MAPE ↓ | | | Acc (10%) ↑ | | |
|---|---|---|---|---|---|---|
| | GNN | Llama3-8B | Mistral-7B | GNN | Llama3-8B | Mistral-7B |
| cpu-openppl-fp32 | **10.48** | 12.57 | 12.22 | **58.94** | 54.91 | 56.26 |
| hi3559A-nnie11-int8 | 7.55 | 6.24 | **5.38** | 73.19 | 80.72 | **88.15** |
| gpu-T4-trt7.1-fp32 | **9.32** | 10.00 | 9.69 | 60.87 | 56.52 | 58.74 |
| gpu-T4-trt7.1-int8 | 18.10 | 15.17 | **14.05** | 27.90 | **47.85** | 46.78 |
| gpu-P4-trt7.1-fp32 | **9.75** | 10.81 | 9.91 | 60.97 | 53.58 | 58.89 |
| gpu-P4-trt7.1-int8 | 13.75 | 12.55 | **12.05** | 36.68 | **48.93** | 48.83 |
| hi3519A-nnie12-int8 | 7.13 | 6.94 | **5.96** | 77.53 | 81.01 | **85.02** |
| atlas300-acl-fp16 | 14.41 | 11.38 | **9.47** | 47.76 | 59.62 | **68.05** |
| mul270-neuware-int8 | **26.18** | 26.88 | 28.31 | 21.61 | 30.77 | **33.70** |
| **Average** | 12.96 | 12.50 | **11.89** | 51.72 | 57.10 | **60.49** |

els, enabling the LLM to process and predict performance metrics more effectively.

## 5.2. Experiment: Effect of Pre-training Strategy

This study assesses the impacts of the GNN pre-training and the graph-to-text adaptation. The hypothesis driving this experiment is that pre-training can provide foundational knowledge that aids in subsequent performance prediction tasks. In this study, we leverage the performance prediction fine-tuning dataset as mentioned in Section 4.3. We used the Llama3-8B as the base LLM, optimizing with a learning rate of 0.0001 for the LLM and 0.001 for the GNN. The training was conducted over 10 epochs for 3 times.

**Result**: The configuration with graph-text adaptation (LLM$_{PRE}$) and pre-trained GNN initialization (GNN$_{PRE}$) significantly outperforms other setups as shown in Table 2. This validates our hypothesis that initial knowledge acquisition through auxiliary tasks can substantially enhance the model's ability to predict performance metrics accurately. Interestingly, GNN pre-trained alone performs worse than randomly initialized GNN. We believe that randomly initialized GNN weights prevent overfitting to pre-existing biases, encouraging the LLM to learn more generalized and robust features during training.

## 5.3. Experiment: Comparison with State-of-the-Art GNN

To rigorously evaluate our proposed architecture, we conducted a comparative analysis against the established GNN baseline (Liu et al., 2022) model across the multi-platform performance prediction fine-tuning dataset as mentioned in Section 4.3. This comparison is crucial to validate the enhancements offered by our approach, particularly in terms of accuracy. In this experiment, we used two variants of our

model: one with Llama3-8B and one with Mistral-7B as the base LLM, both utilizing GNN pre-training and graph-text adaptation.

**Settings:** The baseline GNN model was utilized with no architectural modifications as described in its original implementation. For both of our models, we used the Adam optimizer with a learning rate of 0.0001 for the LLM and 0.001 for the GNN, across 10 training epochs conducted three times.

**Results:** According to the results shown in Table 3, both variants of our model with the pre-training strategy outperformed the GNN baseline. Notably, our model with the Mistral-7B base LLM outperforms the baseline by approximately 8.26% reduction in MAPE and 16.96% (8.8 percentage-point) increase in Acc (10%). These results highlight the critical impact of the effective model representation and the pre-training strategy on the performance prediction capabilities of LLMs.

Additionally, the results show that the choice of LLM significantly affects performance prediction accuracy. For instance, our model with the Mistral-7B base LLM consistently outperforms across various platforms, demonstrating the importance of model selection in achieving higher accuracy.

## 5.4. Experiment: Adaptation

This experiment assesses the real-world adaptability of our model to new hardware environments, particularly under conditions of limited training data. Our comparative analysis involved three models: a standard GNN baseline and two variants of our model, one with and one without both GNN pre-training and graph-text adaptation. Both variants of our model employ Llama3-8B as the base LLM, consistent with the settings described in section 5.3. Each model

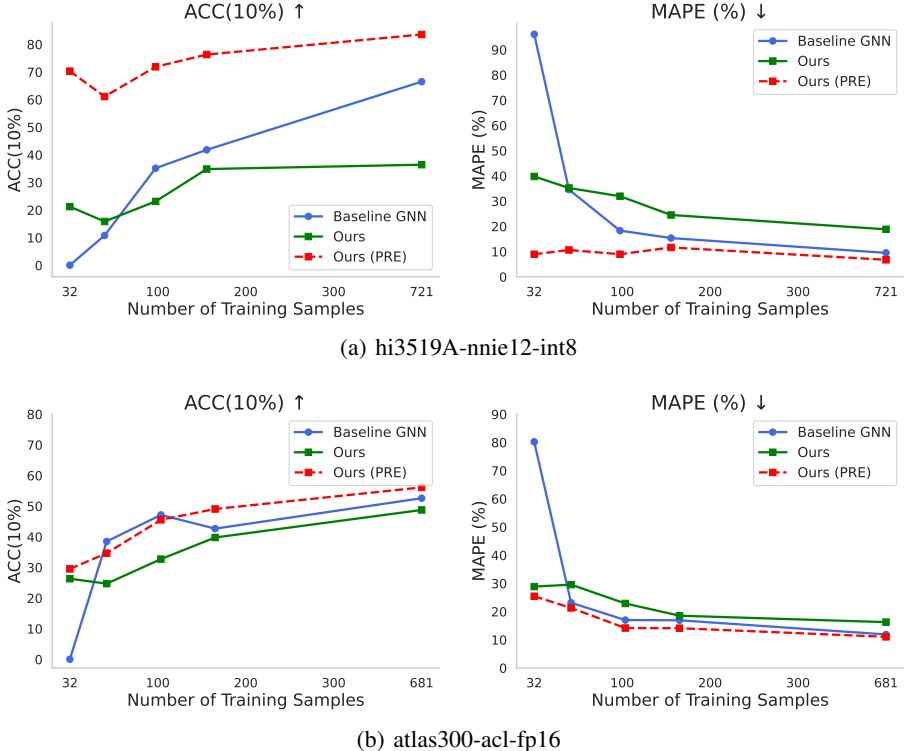

(a) hi3519A-nnie12-int8

(b) atlas300-acl-fp16

Figure 3: Adaptability experiment demonstrating model transfer ability across different hardware platforms. We compared three models: a GNN baseline and two variants of our model-Llama3-8B (with and without the structured pre-training). Each model was trained on eight hardware configurations, followed by a transfer of learned weights to fine-tune on a new unseeen hardware platform (hi3519A-nnie12-int8 or atlas300-acl-fp16) with a varying number of training samples. Our model with the structured pre-training outperformed both the GNN baseline and our model variant without the structured pre-training.

was trained across eight distinct hardware platforms for ten epochs, after which the learned weights were transferred to additional, unseen hardware platforms for further training for three epochs.

The results, illustrated in Figure 3, demonstrate the superior adaptability and performance of our enhanced model on new unseeen hardware with sparse training samples. On the hi3519A-nnie12-int8 and atlas300-acl-fp16 platform, our model equipped with the structured pre-training achieves 70% and 29% Acc(10%) respectively, while GNN achieves 0%, when training on just 32 samples. The results also highlight the importance of our structured pre-training strategy, increasing the accuracy of the LLM-GNN model by up to 50 percentage-point. Notice that without the structured pre-training, the LLM-GNN model even underperformed the GNN baseline in some scenarios. These results underscore the critical roles of both LLMs and our structured-pre-training strategy in enhancing model adaptability, proving essential for the deployment of learned performance modeling in dynamic real-world applications.

# 6. Discussion and Conclusion

This paper has investigated the integration of GNNs and LLMs to enhance the accuracy and adaptability of DL performance prediction. Our empirical evaluations have demonstrated that graph embeddings, derived from GNNs, are more effective inputs for LLMs than traditional text-based representations, leading to significant improvements in both accuracy and computational efficiency. Additionally, we have proposed a structured pre-training strategy that enables model adaptation to new hardware environments with minimal retraining, further enhancing the practicality and efficacy of our approach. We believe that our research offers a promising direction for advancing the field of DL performance prediction and its applications in various stages of the DL system stack.

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

# A. Appendix

### A.1. Environment setup

All experiments were conducted on hardware featuring AMD EPYC 7402 processors with two sockets (24 cores per socket), 512 GB DDR4-3200 RAM, and a 4 x NVIDIA A100 GPU with 40 GB HBM. Our software environment included Python libraries such as PyTorch 2.2.1, torch-geometric 2.5.3, transformers 4.41.0, and peft 0.10.1, running on CUDA version 12.1.

### A.2. Evaluation Metrics

To assess the accuracy of our performance prediction models, we use the following two primary metrics:
**Mean Absolute Percentage Error (MAPE)**: This metric

quantifies the average of the absolute percentage differences between each predicted value and its corresponding actual value. It is defined mathematically as:

$$\text{MAPE} = \frac{1}{n} \sum_{i=1}^{n} \left| \frac{y_i - y_i'}{y_i} \right| \times 100\%$$

Here, $y_i$ represents the actual value and $y_i'$ represents the predicted value. MAPE is a non-negative number, where a smaller value indicates a more accurate model.

**Accuracy within a delta threshold (ACC($\delta$)):** This metric measures the percentage of predicted values that are within a specified percentage ($\delta$) of the actual values. It is defined as:

$$\text{ACC}(\delta) = \frac{1}{n} \sum_{i=1}^{n} \text{pos} \left( \delta - \left| \frac{y_i - y_i'}{y_i} \right| \right) \times 100\%$$

where $\text{pos}(z)$ is a function that returns 1 if $z \leq 0$ and 0 otherwise. A higher ACC($\delta$) value reflects better predictive performance of the model.

These metrics are widely used in previous works(Panner Selvam & Brorsson; Yi et al., 2023; Liu et al., 2022), providing a reliable means to compare the effectiveness of different models in terms of both overall error magnitude and the proportion of predictions within acceptable error thresholds.

### A.3. Datasets

We leverages the NNLQP Multi-Platform dataset (Liu et al., 2022), renowned for its diversity in DL models and hardware configurations. The dataset contains ten different model families, including ResNets, EfficientNets, MobileNets, and GoogleNets, spanning various computational architectures from multiple GPUs to NPUs and FPGAs. This dataset is ideal for assessing our proposed architecture due to its extensive collection of performance metrics across varied hardware platforms. Despite its comprehensive nature, the NNLQP dataset was originally tailored for GNN-based performance prediction models. To align it with our LLM-focused methodology, we have restructured the dataset into a format conducive to LLM processing.

#### A.3.1. GNN PRE-TRAINING DATASET

We transformed the 20K ONNX models into a node feature matrix and adjacency matrix, then converted these into the PyTorch Geometric data format (PyG), as detailed in Section 4.1, for GNN pre-training (Section 4.3).

#### A.3.2. GRAPH TO TEXT DATASET

For graph-text adaptation, we structured the dataset into $\{(G, Q, A)\}$ format: $G$ represents the DL model's graph structure in PyG format, $Q$ is a textual prompt (*Summarise*

*the graph*), and $A$ is the summary of DL architecture, which is the response from the LLM. Similarly, we generated a graph-to-text dataset for the entire NNLQP dataset.

---

**Graph to Text: Sample Prompt**

Q: Summarise the graph

---

A: The graph contains 42 nodes and 40 edges. The batch size is 8. The graph has 12.1 GFLOPs, 0.33 billion parameters, and 0.4 billion MACs.
Layer Stats Summary:
Layer: convolution, Implemented: 20 times, FLOPs: 12.5 billion, MACs: 32.3 million, Parameters: 35.8 million
Layer: relu, Implemented: 17 times, FLOPs: 0, MACs: 0, Parameters: 3.3 million
Layer: max pooling, Implemented: 1 times, FLOPs: 0, MACs: 0, Parameters: 335.6 thousand Layer: addition, Implemented: 1 times, FLOPs: 0, MACs: 0, Parameters: 865

---

#### A.3.3. PERFORM PREDICTION FINE-TUNING DATASET

For general performance prediction tasks, $Q$ queries the LLM along with $G$ to predict inference times. The response $A$ is the predicted inference latency, directly corresponding to the LLM's output.

### A.4. GNN Pre-training Hyper-parameters

### A.5. Limitations and Future Work

While our model effectively leverages static prompting to enhance performance prediction, exploring diverse prompting strategies could further optimize its adaptability and effectiveness across various scenarios.

Future research will explore several avenues to enhance the current model's robustness and applicability. We plan to extend our methodology to additional DL performance datasets such as TPU Graphs(Phothilimthana et al., 2023), allowing us to validate and refine our approach across a wider range of network architectures and operational environments.

### Additional Experiment: Graph Embedding Projection

This experiment investigates the effectiveness of different graph embedding projection techniques, essential for communicating the graph structural information from the GNN encoder to the LLM. It allow gradient flow from the LLM back to the GNN encoder, thereby enhancing the learning feedback loop.

Table 4: Hyper-parameters used for GNN pre-training.

| Hyperparameter | Value |
|---|---|
| Number of Hidden Units | 1024 |
| Number of Features | 44 |
| Number of Layers | 5 |
| Learning Rate (lr) | 0.0005 |
| Weight Decay | 0.00 |
| Mask Rate | 0.5 |
| Drop Edge Rate | 0.0 |
| Maximum Epochs | 500 |
| Encoder Type | GIN |
| Decoder Type | GIN |
| Activation Function | PReLU |
| Loss Function | SCE |
| Use of Scheduler | No |
| Batch Size | 128 |
| Alpha_l | 2 |
| Replace Rate | 0.1 |
| Normalization Type | BatchNorm |
| Optimizer | Adam |
| Input Dropout | 0.2 |
| Attention Dropout | 0.1 |

Table 5: Performance comparison for Single vs. Multi Embedding -Projection methods

| Type | MAPE $\downarrow$ | ACC(10%) $\uparrow$ | TTT $\downarrow$ | Max Token Length |
|---|---|---|---|---|
| Single Proj. | **12.61** | **52.83** | **0.23** | 512 |
| Multi Proj. | 13.66 | 48.50 | 2.32 | 2048 |

**Setting**: For this experiment, we utilized the same dataset and the same Llama3-8B and GIN encoder as DL representation experiment explained in Section 5.1. Each model was trained over 10 epochs 3 times to ensure stability and convergence of results.

**Results**: As result shown in Table 5, our proposed architecture, the single projection technique where the DL graph is projected as a single input embedding to LLM ($g_1$) demonstrated superior performance compared to the multi-projection method, which attempts to capture the graph structure as multiple embeddings from $g_1$ to $g_{H_{GNN}}$. This finding suggests that maintaining a focused, singular projection of graph features into the LLM not only preserves essential structural details but also enhances computational efficiency. This single embedding approach resulted in MAPE, higher ACC(10%), and reduced TTT. These results validate the importance of optimizing graph projection methods to enhance the interplay between GNN encodings and LLM capabilities for performance prediction tasks.

