# OpenReview forum: "Can LLMs Enhance Performance Prediction for Deep Learning Models?"
_ICML.cc/2024/Workshop/WANT — WANT@ICML 2024 Oral_

### Official Review · Reviewer_QfET · 2024-06-12
**A promising approach that needs a more rigorous comparison**

**Confidence:** 3

**Summary:**

Authors propose an approach which combines GNN and LLM to predict the performance prediction for DL models. The GNN is used to get embeddings riched with information about DL model and LLM is used to predict the performance. The approach is tested against several baselines and across various datasets.

**Strengths:**

1. A novel pre-training strategy is introduced.
2. The approach is tested on a variety of datasets.
3. The approach is compared to the existing baselines.

**Weaknesses:**

1. There is no discussion why the baseline GNN outperforms the proposed approach on the datasets with suffix "fp32". (Section 5.3)
2. The proposed approach utilizes the pre-training strategy, but there is no information on the pre-training the GNN baseline. If the baseline was not pre-trained, the comparison is unfair. (Section 5.3)
3. The choice of datasets for comparison in section 5.4 is not explained. The results lack statistical significane test analysis.

---

### Official Review · Reviewer_QorJ · 2024-06-13
**Review of "Can LLMs Enhance Performance Prediction for Deep Learning Models?" paper**

**Confidence:** 5

**Summary:**

The study highlights the potential of integrating GNNs and LLMs to enhance DL model performance prediction, particularly in terms of adaptability to new environments and workloads. Current methods, though accurate, are slow to adapt to new hardware environments or new types of workloads so authors propose combining Graph Neural Networks (GNNs) and Large Language Models (LLMs) to improve prediction accuracy and adaptability. GNNs are good at capturing the structural information of DL models, which are naturally represented as graphs; LLMs provide generalization and adaptability to various tasks due to their extensive pre-training on large datasets.

**Strengths:**

1) All sections are well-structured and provide enough information
2) The research introduces a novel graph-to-text dataset designed to further research into the integration of GNNs and LLMs. This dataset is valuable for benchmarking and advancing the application of GNN-LLM combinations in graph learning tasks​.

**Weaknesses:**

1) Although the paper introduces a novel dataset for the graph-to-text adaptation, it relies heavily on the NNLQP dataset, which might limit the generalizability of the findings. Expanding the datasets to include more diverse DL models and hardware configurations could provide more robust validation.
2) The model's effectiveness is heavily dependent on the structured pre-training strategy. If this pre-training is not carefully managed or if the initial datasets are not representative enough, the model might not perform as well in real-world scenarios.
3) The experiments primarily focus on a specific set of hardware configurations and DL models. A broader range of experiments covering more diverse scenarios could strengthen the claims made in the paper .

---

### Official Review · Reviewer_1Ajr · 2024-06-14
**Review for Can LLMs Enhance Performance Prediction for Deep Learning Models**

**Confidence:** 2

**Summary:**

The paper presents a method to leverage both GNNs and LLMs to improve the accuracy and adaptability of DL  performance prediction. From empirical evaluation, this method outperforms traditional representations such as high-level text summary and lossless semi-structured text on DL performance prediction. In addition, this paper also proposed a structured pre-training strategy to enable model adaptation to new hardware environments, significantly reducing the need for extensive retraining.

**Strengths:**

The methods are described clearly and the evaluation supports the conclusion

**Weaknesses:**

The limitation of the method is not mentioned much.

---

### Meta-Review · Area_Chair_RHPr · 2024-06-18

**Recommendation:** Accept (Oral)
**Confidence:** 5

**Metareview:**

**Strengths**
- The work introduces multiple novel ideas for performance prediction, including GNN+LLM integration, pre-training strategy, and text-adaption dataset.
- Paper is well structured and ideas intuitively explained.
- Evaluation clearly shows advantage over prior work.

**Weaknesses**
- Limitation of the approach is not well discussed. This would be important to help readers understand the applicability to their scenarios.
- An explanation of why baseline GNN outperforms on `fp32` datasets is needed. This could be related to the limitation issue.

**Summary**
- This is great work to share with the community.

---

### Decision · Program_Chairs · 2024-06-18

**Decision:**

Accept (Oral)

**Comment:**

We thank the authors for their time and contribution to WANT and we are pleased to share that after the reviewing process the paper has been accepted. Congratulations! We encourage the authors to consider reviewers' feedback for the improvement of the camera-ready version. We hope to see you in person at the workshop and brainstorm on efficient training research together!